# Enhancing Consistent Ground Maneuverability by Robot Adaptation to Complex Off-Road Terrains

**Sriram Siva**[1], **Maggie Wigness**[2], **John G. Rogers**[2], and **Hao Zhang**[1]
[1]Colorado School of Mines, Golden, CO 80401, USA
{sivasriram,hzhang}@mines.edu
[2]U.S. Army Research Laboratory, Adelphi, MD 20783, USA
{maggie.b.wigness.civ,john.g.rogers59.civ}@army.mil

**Abstract:** Terrain adaptation is a critical ability for a ground robot to effectively traverse unstructured off-road terrain in real-world field environments such as forests. However, the expected or planned maneuvering behaviors cannot always be accurately executed due to setbacks such as reduced tire pressure. This inconsistency negatively affects the robot's ground maneuverability, and can cause slower traversal time or errors in localization. To address this shortcoming, we propose a novel method for consistent behavior generation that enables a ground robot's actual behaviors to more accurately match expected behaviors while adapting to a variety of complex off-road terrains. Our method learns offset behaviors in a self-supervised fashion to compensate for the inconsistency between the actual and expected behaviors without requiring the explicit modeling of various setbacks. To evaluate the method, we perform extensive experiments using a physical ground robot over diverse complex off-road terrain in real-world field environments. Experimental results show that our method enables a robot to improve its ground maneuverability on complex unstructured off-road terrain with more navigational behavior consistency, and outperforms previous and baseline methods, particularly so on challenging terrain such as that which is seen in forests.

**Keywords:** Robot Learning, Off-road Navigation, Terrain Adaptation

## 1 Introduction

Over the past several years, autonomous ground robots have been increasingly deployed in off-road field environments to address real-world applications, including disaster response, homeland defense, and planetary exploration [1, 2, 3]. Field environments are challenging for ground robots to navigate over because the terrain is unstructured and cannot be fully modeled beforehand, as depicted in Fig. 1. Terrain adaptation, the robot's ability to adjust its behaviors to perceived terrains, is therefore an essential ability to traverse over unstructured terrains [4, 5].

Given its importance, robot terrain adaptation has recently been widely investigated. Previous learning-based methods can be divided into two broad categories: terrain classification and terrain adaptation. The first category uses a robot's exteroceptive and proprioceptive sensory data to classify terrain types and estimate traversability

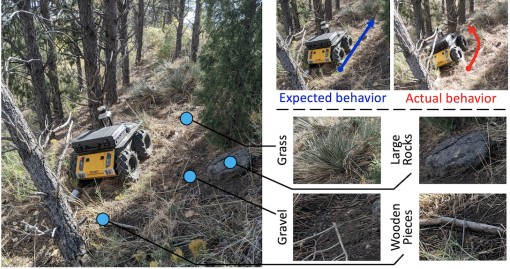

Figure 1: Off-road environments such as forests are unstructured and exhibit a variety of characteristics, including changing terrain types and slopes. When ground robots are deployed in these environments, their actual behaviors often do not match the expected behaviors, e.g., due to wheel slip. The inconsistency often causes slower traversal time or errors in robot state estimation. Therefore, the capability of consistent behavior generation is essential for maneuverability while ground robots navigate over unstructured off-road terrains.

for robot navigation over the terrain [6, 7, 8, 9, 10]. This category also includes techniques that model terrain complexity for navigation planning [5, 11]. The second category of methods focus on directly

5th Conference on Robot Learning (CoRL 2021), London, UK.

generating adaptive behaviors according to terrain in order to successfully complete navigation tasks [5, 12, 13, 14]. Specifically, learning from demonstration (LfD) is widely used to transfer human expertise to robots in order to achieve human-level robot navigational control [15, 14, 16].

However, the expected navigational behaviors generated by previous approaches cannot always be executed accurately by a ground robot when traversing over unstructured off-road terrain, i.e., the actual and expected behaviors may not be consistent. This inconsistency is mainly caused by setbacks [17, 18] that are defined as factors that increase the difficulty for a ground robot to achieve its expected behaviors. Example setbacks include wheel slip, reduced tire pressure, and heavy payload. Existing learning-based methods for robot navigation generally ignore these setbacks, which often leads to the robot not being able to consistently execute the learned behaviors. The challenge of how to learn consistent navigational behaviors in a self-supervised fashion has not been well addressed.

To address this shortcoming, we develop an approach for consistent navigational behavior generation. Our approach learns offset behaviors in a self-supervised fashion, allowing the robot to compensate for the inconsistency between the actual and expected behaviors without explicitly modeling the setbacks, while also adaptively navigating over changing terrain. In addition, our method is able to integrate multi-modal features to characterize terrain and estimate the importance of features to enable terrain-aware ground navigation. This is all implemented in a unified regularized optimization framework with a theoretical convergence guarantee.

The key novelty of this paper is the introduction of a method to enhance consistent ground maneuverability, which advances the state-of-the-art by enabling a ground robot's actual behaviors to accurately match its expected behaviors while adapting to a variety of complex unstructured off-road terrain. The specific novel contributions include:

- We propose a novel mathematical formulation to generate consistent navigational behaviors by learning offset behaviors in a self-supervised fashion. We also introduce new regularization terms to learn important terrain features from multi-sensory observations and fuse them together to improve robustness of robot adaptation to unstructured terrain.

- We propose a new optimization algorithm to address the formulated regularized optimization problem with dependent variables and non-term regularization terms, which holds a theoretical guarantee to effectively converge to the global optimal solution.

As an experimental contribution, we provide a comprehensive performance evaluation of learning-based terrain adaptation methods by designing a set of robot navigation scenarios over a wide variety of individual and complex unstructured off-road terrains.

## 2    Related Work

Related research on robot terrain adaptation can be broadly classified into two categories, including terrain classification and robot adaptation.

Terrain classification methods use sensory data from a robot to classify the terrain. Many earlier methods were developed to address the specific needs of larger vehicles [19, 20] and the classification process was typically performed in a manual or pre-selected fashion. Some methods used a pre-existing terrain map and terrain ruggedness data to achieve high-speed terrain navigation [6, 7, 9, 10, 21, 22]. Learning-based methods are commonly used to classify terrain for navigation. Color-based terrain classification was performed to generate navigational behaviors by labeling obstacles [23]. More recently, semantic segmentation neural network architectures [24] have been successfully used to classify off-road terrain [25, 26] and perform robot navigation tasks accordingly. However, these methods rely on a discrete categorization of terrain types. The methods typically do not characterize complex unstructured off-road terrain well and cannot directly enable robot adaptation in real-world environments, where environments have a wide variety of characteristics (as seen in Fig. 1).

Robot adaptation methods focus on enabling robots to intelligently adapt to various unstructured terrain. The general problem of robot adaptation is commonly studied in robotics [27, 28, 29, 30, 31, 32] using high-level behavior models. Earlier works considered ground speed as the optimization variable and formulated a method for trading progress and velocity with changing terrain characteristics [33, 34]. Learning-based methods for terrain adaptation have gained attention because of their effectiveness and flexibility [35]. Early work addressed terrain adaptation from the perspective of

online learning that updates model parameters in the execution phase [36, 37, 38]. However, these methods lacked the ability to quickly adapt to sudden terrain changes. Accordingly, methods were developed that generate navigational behaviors using open-loop controllers according to the predicted terrain characteristics [39]. Methods to mimic expert controls were also developed aiming to achieve human-level maneuverability [15, 14]. Recently, navigational affordances are learned from experts for off-road navigation [40]. External disturbances were also considered in the learning-based methods [41, 42, 43]. However, these methods often require modeling of robot dynamics or explicit human demonstrations, and cannot generate consistent ground navigation that adapt to diverse terrains.

## 3 Approach

**Notation:** We denote scalars using lowercase italic letters (e.g., $m \in \mathbb{R}$), vectors using boldface lowercase letters (e.g., $\mathbf{m} \in \mathbb{R}^p$), matrices using boldface capital letters, e.g., $\mathbf{M} = \{m_j^i\} \in \mathbb{R}^{p \times q}$ with its $i$-th row and $j$-th column denoted as $\mathbf{m}^i$ and $\mathbf{m}_j$, respectively. We use boldface capital Euler script letters to denote tensors (i.e., 3D matrices), e.g., $\boldsymbol{\mathcal{M}} = \{m_j^{i(k)}\} \in \mathbb{R}^{p \times q \times r}$. Unstacking tensor $\boldsymbol{\mathcal{M}}$ along its height ($p$), width ($q$) and depth ($r$) provides slices of matrices $\mathbf{M}^i \in \mathbb{R}^{q \times r}$, $\mathbf{M}_j \in \mathbb{R}^{p \times r}$ and $\mathbf{M}^{(k)} \in \mathbb{R}^{p \times q}$, respectively [44].

### 3.1 Problem Formulation for Terrain-Aware Navigation

As a robot traverses over terrain, at each time step, we extract multi-modal features from observations acquired from multiple sensors installed on the robot including visual camera, LiDAR, and IMU. We concatenate all features extracted at time step $t$ into a vector and denote it as $\mathbf{x}^{(t)} \in \mathbb{R}^d$, where $d = \sum_{j=1}^m d^j$ and $d^j$ is the dimensionality of the $j$-th feature modality with $m$ as the number of modalities. We stack features extracted from a sequence of consecutive $c$ time steps into a matrix and represent it as a terrain feature instance denoted as $\mathbf{X} = [\mathbf{x}^{(t)}; \ldots; \mathbf{x}^{(t-c)}] \in \mathbb{R}^{d \times c}$. We further denote the set of $n$ feature instances that are obtained as a robot traverses over various terrains, and denote this set as a terrain feature tensor $\boldsymbol{\mathcal{X}} = [\mathbf{X}_1, \ldots, \mathbf{X}_n] \in \mathbb{R}^{d \times n \times c}$.

We use $\mathbf{Y} = [\mathbf{y}_1, \ldots, \mathbf{y}_n] \in \mathbb{R}^{r \times n}$ to denote the robot's expected navigational behaviors associated with $\boldsymbol{\mathcal{X}}$, where $\mathbf{y}_i \in \mathbb{R}^r$ is a vector of $r$ behaviors corresponding to $\mathbf{X}_i$. Behaviors are encoded by the control variables (e.g., linear and angular velocities) that decide the robot's motion at the present time $t$. Due to momentum, a robot often has continuous motion and observations; thus, considering a history of past $c$ observations can provide more information to generate navigational behaviors at the present time. Accordingly, we estimate the robot's behaviors $\mathbf{y}_i$ using $\mathbf{X}_i = [\mathbf{x}_i^{(t)}; \ldots; \mathbf{x}_i^{(t-c)}]$, taking into account the history of $c$ observations. Then, the problem of navigational behavior generation can be formulated as:

$$\min_{\boldsymbol{\mathcal{W}}} \|\boldsymbol{\mathcal{W}} \otimes_3 \boldsymbol{\mathcal{X}} - \mathbf{Y}\|_F^2 + \lambda_1 \|\boldsymbol{\mathcal{W}}\|_M \qquad (1)$$

where $\boldsymbol{\mathcal{W}} \in \mathbb{R}^{d \times r \times c}$ is a weight tensor used to encode the importance of each element in $\boldsymbol{\mathcal{X}}$ towards estimating navigational behaviors. Each tensor element $w_j^{i(k)} \in \boldsymbol{\mathcal{W}}$ denotes the weight of the $i$-th terrain feature from the $k$-th past time step to recognize the $j$-th behavior type. The operator $\otimes$ denotes the tensor product and $\otimes_3$ is defined as the tensor product that performs the sum of mode-3 multiplication [44] between feature tensor $\boldsymbol{\mathcal{X}}$ and $\boldsymbol{\mathcal{W}}$. In Eq. (1), the tensor product $\otimes_3$ takes each terrain feature instance $\mathbf{X}_i \in \boldsymbol{\mathcal{X}}$, and multiplies it with the weight tensor $\boldsymbol{\mathcal{W}}$.

The first term in Eq. (1) is a loss function that encodes the error of using terrain features in $\boldsymbol{\mathcal{X}}$ to estimate the robot behaviors, through the learning model parameterized by $\boldsymbol{\mathcal{W}}$. Furthermore, our loss function encodes the non-linear nature of robot navigational behavior generation as a linear function of terrain features $\boldsymbol{\mathcal{X}}$. This can be achieved because: i) for short periods of time (i.e., $c = [1, 30]$), the dynamics of the robot do not dramatically change [45, 46] and thus the non-linearity in the robot is not severe, and ii) with high-dimensional features (e.g., $d > 10000$), learning non-linear tasks can be lifted to a linear space [47, 48]. The second term in Eq. (1) is a regularization term named the feature modality norm and is mathematically defined as:

$$\|\boldsymbol{\mathcal{W}}\|_M = \sum_{i=1}^m \|\mathbf{W}^i\|_F = \sum_{i=1}^m \sqrt{\sum_{j=1}^r \sum_{k=1}^c (\mathbf{w}_j^{i(k)})^\top (\mathbf{w}_j^{i(k)})} \qquad (2)$$

where $\|.\|_F$ is the Frobenius norm and $\mathbf{W}^i \in \mathbb{R}^{b \times c}$ is the slice of the matrix obtained by unstacking the weight tensor $\mathcal{W}$ along its height $d$. The feature modality norm groups together weights within a feature modality and enforces sparsity among different modalities, thus, identifying the most descriptive features for behavior generation. This is a critical capability for ground robot navigation since different features typically capture different characteristics of the unstructured terrains (e.g., color, slope, and roughness), and have different effects toward generating navigational behaviors. The trade-off hyperparameter $\lambda_1$ in Eq. (1) is used to balance the loss and the regularization term.

The problem formulation in Eq. (1) allows ground robots to adapt their navigational behaviors according to different terrain features. However, due to setbacks that reduce the effectiveness of robot navigation, such as wheel slip, heavy payload, and reduced tire pressure [49], the robot's actual behaviors may not match the expected behaviors.

## 3.2 Consistent Navigational Behavior Generation

The key novelty focuses on a principled method for ground robots to generate consistent navigational behaviors that adapt to unstructured terrain. As illustrated by Fig. 2, besides generating terrain-aware expected behaviors, our method monitors the difference between the actual and expected navigational behaviors caused by setbacks, and computes an offset to reduce the difference. This allows our approach to achieve consistent robot behaviors without the requirement of explicitly modeling all of the setbacks.

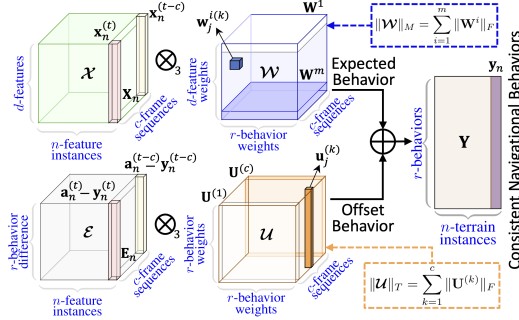

Figure 2: Overview of our proposed approach.

Mathematically, we denote the actual behaviors as $\mathbf{A} = [\mathbf{a}_1, \ldots, \mathbf{a}_n] \in \mathbb{R}^{r \times n}$, where $\mathbf{a}_i \in \mathbb{R}^r$ denotes the actual behaviors executed by the robot when observing the $i$-th terrain feature instance $\mathbf{X}_i$. The actual robot behaviors can be estimated using a pose estimation technique, e.g., based on SLAM or visual odometry [50]. Setbacks can cause the robot's actual behaviors to not match its expected behaviors. This difference in navigational behaviors over the past $c$-time steps is computed as $\mathbf{E} = [(\mathbf{a}^{(t)} - \mathbf{y}^{(t)}); \ldots; (\mathbf{a}^{(t-c)} - \mathbf{y}^{(t-c)})] \in \mathbb{R}^{r \times c}$ and the behavior differences for all the instances in $\mathcal{X}$ is denoted as a behavior difference tensor $\mathcal{E} = [\mathbf{E}_1, \ldots, \mathbf{E}_n] \in \mathbb{R}^{r \times n \times c}$.

We then introduce a loss function to encode consistent behavior generation as follows:

$$\min_{\mathcal{U}, \mathcal{W}} \|\mathcal{W} \otimes_3 \mathcal{X} + \mathcal{U} \otimes_3 \mathcal{E} - \mathbf{A}\|_F^2 + \lambda_1 \|\mathcal{W}\|_M \tag{3}$$

where $\mathcal{U} = [\mathbf{U}^1, \ldots, \mathbf{U}^r] \in \mathbb{R}^{r \times r \times c}$ is a weight tensor with $\mathbf{U}^j \in \mathbb{R}^{r \times c}$ indicates the importance of behavior differences $\mathbf{E}$ towards generating the $j$-th offset behaviors. Using data from a history of $c$ time steps allows our method to consider inertia (i.e., resistance to change in behaviors) during navigation. Mathematically, this implies that $\mathbf{U}^{(k)}, k = 1, \ldots, c$; has non-zero elements as opposed to each element in $\mathbf{U}^{(k)}$ being 1 at $k = 1$ and 0 at $k \neq 1$ when robot inertia is not considered.

The loss function in Eq. (3) models the actual behavior by considering both terrain features $\mathcal{X}$ and the behavior differences $\mathcal{E}$ to achieve consistent navigational behaviors. Because of inertia, historical data from different past time steps may contribute differently towards generating offset behaviors (e.g., a heavier robot with larger inertia often needs to consider a longer history). Thus, we propose a new regularization term to explore which time steps in the historical data are more important for generating the offset behaviors. We name this regularization term the temporal norm, which is expressed by:

$$\|\mathcal{U}\|_T = \sum_{k=1}^{c} \|\mathbf{U}^{(k)}\|_F = \sum_{k=1}^{c} \sqrt{\sum_{j=1}^{r} \|\mathbf{u}_j^{(k)}\|_2^2} \tag{4}$$

where $\mathbf{U}^{(k)} = [\mathbf{u}_1^{(k)}, \ldots, \mathbf{u}_r^{(k)}] \in \mathbb{R}^{r \times r}$ is the weight matrix, and $\mathbf{u}_j^{(k)}$ indicates the importance of the $j$-th behavior difference $\mathbf{a}_j^{(k)} - \mathbf{y}_j^{(k)}$ from $k$-the past time step towards generating offset behaviors. This norm groups together weights for the vector of behavior differences at each time step, and enforces sparsity between weights at different time steps to identify the most important time steps.

---
**Algorithm 1:** Our algorithm to solve the regularized optimization problem in Eq. (5).

    **Input**    : $\mathcal{X} \in \mathbb{R}^{d \times n \times c}$, $\mathbf{A} \in \mathbb{R}^{r \times n}$, and $\mathcal{E} \in \mathbb{R}^{r \times n \times c}$
    **Output**  :The weight tensors $\mathcal{W} \in \mathbb{R}^{d \times r \times c}$ and $\mathcal{U} \in \mathbb{R}^{r \times r \times c}$
**1** Initialize $\mathcal{W} \in \mathbb{R}^{d \times r \times c}$ and $\mathcal{U} \in \mathbb{R}^{r \times r \times c}$;
**2** **while** *not converge* **do**
**3**      Calculate each block diagonal matrix $\mathbf{Q}^{(k)}$ with $i$-th diagonal block given as $\frac{1}{2\|\mathbf{W}^{i(k)}\|_F}\mathbf{I}_{d^i}$;
**4**      Compute the matrices $\mathbf{W}^{(k)}$ according to Eq. (8);
**5**      Calculate the block diagonal matrix $\mathbf{P}$ with the $k$-th diagonal block as $\frac{1}{2\|\mathbf{U}^{(k)}\|_F}\mathbf{I}_r$;
**6**      Compute the matrices $\mathbf{U}^{(k)}$ according to Eq. (10);
**7** **return:** $\mathcal{W} \in \mathbb{R}^{d \times r \times c}$ and $\mathcal{U} \in \mathbb{R}^{r \times r \times c}$
---

Using both norms to generate consistent robot navigational behaviors while identifying important feature modalities and historical time steps, the final objective function becomes:

$$\min_{\mathcal{U}, \mathcal{W}} \|\mathcal{W} \otimes_3 \mathcal{X} + \mathcal{U} \otimes_3 \mathcal{E} - \mathbf{A}\|_F^2 + \lambda_1 \|\mathcal{W}\|_M + \lambda_2 \|\mathcal{U}\|_T \tag{5}$$

where $\lambda_1 \geq 0$ and $\lambda_2 \geq 0$ are trade-off hyper-parameters.

After computing the optimal values of the weight tensors $\mathcal{W}$ and $\mathcal{U}$ according to Algorithm 1, in the training phase, a robot can apply our self-reflective terrain-aware adaptation method to generate consistent navigational behaviors during execution. At each time step $t_0$ in the execution phase, the robot extracts multi-modal features $\mathbf{X}_{t_0}$ from observations obtained from its on-board sensors over the past $c$-time steps. Then, the robot also estimates the corresponding actual behaviors (measured using a pose estimation technique based upon SLAM or visual odometry [50]) and computes the matrix of behavior differences at time $t_0$ as $\mathbf{E}_{t_0}$. Then our approach can be used by the robot to generate consistent actual navigational behaviors as:

$$\mathbf{y} = \mathcal{W} \otimes_3 \mathbf{X}_{t_0} + \mathcal{U} \otimes_3 \mathbf{E}_{t_0} \tag{6}$$

The first term in Eq. (6) generates the expected navigational behaviors according to the terrain it traverses, which allows a ground robot to adapt its navigational behaviors to unstructured terrains. The second term in Eq. (6) provides offset behaviors based on monitoring the difference between actual and expected behaviors in order to compensate for the setbacks.

## 4 Optimization Algorithm

The optimization problem in Eq. (5) is challenging to solve because the regularization terms are not smooth and because the objective function includes dependent variables. Thus, we derive a new iterative optimization algorithm to obtain the optimal solution to Eq. (5). This algorithm is shown in Algorithm 1, which provides an alternating minimization algorithm that alternatively updates the parameter tensors in each iteration until convergence.

To solve for the optimal weight tensor $\mathcal{W}$, we minimize Eq. (5) with respect to $\mathbf{W}^{(k)}$ resulting in:

$$2\mathbf{X}^{(k)}(\mathbf{X}^{(k)})^\top \mathbf{W}^{(k)} - 2\mathcal{X} \otimes_3 \mathbf{A} + 2(\mathcal{U} \otimes_3 \mathcal{E})^\top \otimes_3 \mathcal{X} + \lambda_1 \mathbf{Q}^{(k)} \mathbf{W}^{(k)} = 0 \tag{7}$$

where each $\mathbf{Q}^{(k)} \in \mathbb{R}^{d_i \times d_i}$ is a diagonal matrix with the $i$-th diagonal block computed by $\frac{1}{2\|\mathbf{W}^{i(k)}\|_F}\mathbf{I}_{d^i}$ and $\mathbf{I}_{d^i}$ is an identity matrix. Then, we compute each $\mathbf{W}^{(k)}$ as:

$$\mathbf{W}^{(k)} = \left(2\mathbf{X}^{(k)}(\mathbf{X}^{(k)})^\top + \lambda_1 \mathbf{Q}^{(k)}\right)^{-1}\left(2\mathcal{X} \otimes_3 \mathbf{A} - 2(\mathcal{U} \otimes_3 \mathcal{E})^\top \otimes_3 \mathcal{X}\right) \tag{8}$$

Because each of the block-diagonal matrices $\mathbf{Q}^{(k)}$ are dependent on $\mathcal{W}$ and also each slice of matrix $\mathbf{W}^{(k)}$ is dependent on corresponding $\mathbf{Q}^{(k)}$, an iterative algorithm is required to compute them.

To compute the optimal $\mathcal{U}$, we calculate the derivative of the objective function in Eq. (5) with respect to $\mathbf{U}^{(k)}$ and set the equation to zero as:

$$2\mathbf{E}^{(k)}(\mathbf{E}^{(k)})^\top \mathbf{U}^{(k)} - 2\mathbf{A} \otimes_3 \mathcal{E} + 2(\mathcal{W} \otimes_3 \mathcal{X})^\top \otimes \mathcal{E} + \lambda_2 \mathbf{P}\mathbf{U}^{(k)} = 0 \tag{9}$$

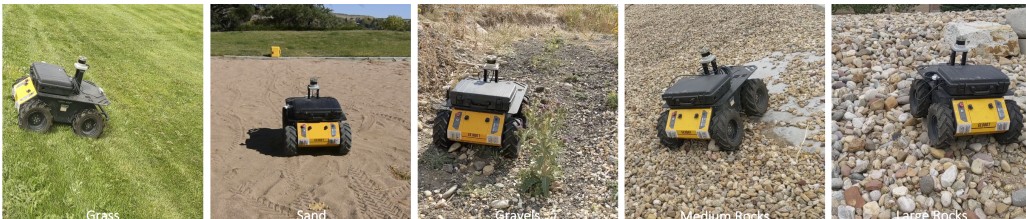

Figure 3: Individual types of unstructured terrain used in the experiments.

where $\mathbf{P} \in \mathbb{R}^{r \times r}$ is a diagonal matrix with the $k$-th diagonal block as $\frac{1}{2\|\mathbf{U}^{(k)}\|_F}\mathbf{I}_r$, with $\mathbf{I}_r$ being an identity matrix. Then, we compute each $\mathbf{U}^{(k)}$ in a closed-form solution as:

$$\mathbf{U}^{(k)} = \left(2\mathbf{E}^{(k)}(\mathbf{E}^{(k)})^\top + \lambda_2\mathbf{P}\right)^{-1}\left(2\mathbf{A} \otimes_3 \boldsymbol{\mathcal{E}} - 2(\boldsymbol{\mathcal{W}} \otimes_3 \boldsymbol{\mathcal{X}})^\top \otimes \boldsymbol{\mathcal{E}}\right) \tag{10}$$

This $\boldsymbol{\mathcal{U}}$ is then used to calculate $\boldsymbol{\mathcal{W}}$ in the next iteration. Similar to $\mathbf{Q}^{(k)}$ and $\boldsymbol{\mathcal{W}}$, $\mathbf{P}$ and $\boldsymbol{\mathcal{U}}$ are interdependent. Thus, we develop an iterative algorithm to solve the formulated optimization problem, which is described in Algorithm 1.

**Convergence.** Algorithm 1 is guaranteed to converge to the global optimal solution to the formulated regularized optimization problem in Eq. (5). The proof is provided in the supplementary material.

**Complexity.** As the optimization problem in Eq. (5) is convex, Algorithm 1 converges fast (e.g., within tens of iterations only). In each iteration, computing Steps 3 and 5 is trivial. Steps 4 and 6 can be computed by solving a system of linear equations with quadratic complexity.

## 5    Experiments

We utilize the Clearpath Husky robot in our field experiments. The robot is equipped with an Intel Reasense D435 color-depth camera and an Ouster OS1-64 LiDAR. The robot also has a variety of sensors to measure its internal states, including IMU readings, wheel odometry, motor speed, and battery status. Linear interpolation is used to get a steady 30 Hz frame rate from all sensors.

To represent unstructured terrain, we implement multiple visual features extracted from color images to describe different terrain characteristics, including Histogram of Oriented Gradients (HOG) [51] to describe the shape and Local Binary Patterns (LBP) [52] to describe texture. We also compute an elevation map [53] from LiDAR data to represent robot-centric grid-wise elevation of the terrain. During training, expected navigational behaviors are provided by remote control to navigate the robot over unstructured terrains as fast as possible while maintaining safety. In the training and execution phases, actual behaviors are estimated from LiDAR-based SLAM [50] as robot pose. We use a sequence of 15 frames (i.e., $c = 15$), and $\lambda_1 = 0.1$ and $\lambda_2 = 10$ for all experiments. More details on our implementation and training/testing procedures are provided in the supplementary material.

We compare our approach with several previous state-of-the-art learning-based robot navigation techniques, including Learning from Demonstration (LfD) for robot navigation [16], multi-modal LfD (MfD) [54], and Terrain Representation and Apprenticeship Learning (TRAL) [15]. To quantitatively evaluate the performance of robot navigation, we use four metrics:

- *Failure Rate (FR)*: This metric is defined as the number of times the robot fails to complete the navigation task across a set of experimental trials. If a robot flips or is stopped by a terrain obstacle, it is considered a failure. Lower values of FR indicate better performance.

Table 1: Quantitative results based on ten runs for scenarios when the robot traverses over *individual types of unstructured terrain*. Successful runs (with no failures) are used to calculate the metrics of traversal time, inconsistency and jerkiness.

| Terrain | Failure Rate (/10) | | | | Traversal Time (s) | | | | Inconsistency | | | | Jerkiness (m/s³) | | | |
|---|---|---|---|---|---|---|---|---|---|---|---|---|---|---|---|---|
| | LfD | MfD | TRAL | **Ours** | LfD | MfD | TRAL | **Ours** | LfD | MfD | TRAL | **Ours** | LfD | MfD | TRAL | **Ours** |
| Grass | 0 | 0 | 0 | **0** | 17.9 | 18.2 | **17.4** | 17.5 | 2.82 | 3.06 | **1.84** | 2.11 | 79.59 | 81.42 | **75.18** | 76.50 |
| Sand | 0 | 1 | 0 | **0** | 15.3 | **12.7** | 15.9 | 13.1 | 4.72 | 4.62 | 4.67 | **4.55** | 71.14 | 73.74 | 65.32 | **64.22** |
| Gravel | 0 | 0 | 0 | **0** | 22.9 | 24.1 | 20.4 | **20.2** | 4.12 | 4.63 | 3.81 | **3.04** | 44.95 | 48.27 | 40.26 | **39.48** |
| M.R | 1 | 3 | 0 | **0** | 33.2 | 36.9 | 29.4 | **28.4** | 7.92 | 9.59 | 4.21 | **2.41** | 141.48 | 144.22 | 113.34 | **111.18** |
| L.R | 6 | 6 | 2 | **1** | 57.8 | **55.3** | 63.4 | 60.9 | 24.79 | 28.50 | 9.51 | **7.84** | 52.55 | 54.30 | 49.50 | **48.36** |

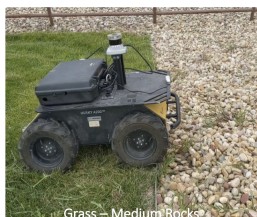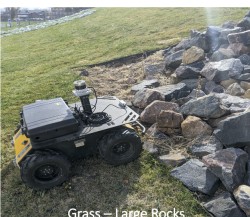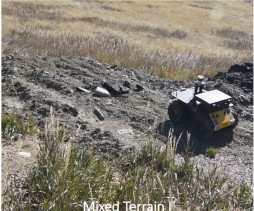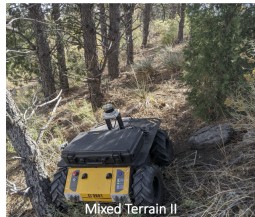

Figure 4: Complex unstructured off-road terrains used in the experiments.

- *Traversal Time (TT)*: This metric is defined as the time taken to complete the navigation task over given terrain. Smaller values of TT indicate better performance.

- *Inconsistency*: This metric is defined as the error between the expected behavior and the actual behavior in terms of robot poses (linear position and angular). Lower values of inconsistency indicate better performance.

- *Jerkiness*: This metric is defined as the average sum of the acceleration derivatives along all axes, with lower values indicating better performance. Jerkiness indicates how smooth a robot can traverse over a terrain. Because state estimation and SLAM methods (e.g., based on Kalman filters) may assume smooth robot motions, jerkiness is a useful metric.

## 5.1 Navigating over Individual Types of Unstructured Terrain

In this set of experiments, the robot navigates over individual terrain using off-road tracks. Each track is made up of one type of terrain and is approximately ten meters long. Five types of terrain are used in our experiments, which are illustrated in Fig. 3. Our approach is trained on data collected while the robot is manually controlled by an expert to traverse the terrain. Then, the learned model is deployed on the robot to autonomously navigate over the terrain. Evaluation metrics for each method are computed across ten trials on each type of individual terrain track.

The quantitative results achieved by our approach and the comparison to other methods are presented in Table 1. For simple individual terrains, such as grass and gravel, all methods allow the robot to successfully traverse over the terrain. However, for more challenging terrain, especially large rocks, both LfD and MfD have a high failure rate, whereas, TRAL and our approach have a low failure rate. Our approach only has one failure over the difficult large-rock terrain and outperforms other tested methods. The presented traversal time is computed by averaging the traversal time across all successful runs, i.e., it excludes the failed trials captured by the FR metric. It is observed that all methods have a similar traversal time. In successful runs, both LfD methods show less traversal time compared to other methods over rocky terrain, although they also have a much higher failure rate. Thus, the emphasis on high-speed traversal used by both the LfD methods produces an unreliable system when the robot traverses unstructured off-road terrain in real-world field environments.

Table 1 also presents the quantitative results for the inconsistency and jerkiness metrics. We observe that both LfD and MfD methods do not perform well and have higher values of inconsistency over individual types of terrain, especially on the large-rock terrain. TRAL has lower inconsistency and performs the best over the grass terrain. Our proposed method outperforms the previous approaches and obtains the lowest averaged inconsistency value. Finally, we also evaluate the tested methods using the jerkiness metric. An observation is that the medium-rock terrain causes the largest jerkiness measure. This is because medium-rocks terrain as compared to the large-rocks produce much more vibrations for even slow maneuvers.

Table 2: Quantitative results for scenarios when the robot traverses over *complex unstructured off-road terrain* shown in Fig. 4.

| Terrain | Failure Rate (/10) | | | | Traversal Time (s) | | | | Inconsistency | | | | Jerkiness (m/s³) | | | |
|---|---|---|---|---|---|---|---|---|---|---|---|---|---|---|---|---|
| | LfD | MLfD | TRAL | **Ours** | LfD | MLfD | TRAL | **Ours** | LfD | MfD | TRAL | **Ours** | LfD | MfD | TRAL | **Ours** |
| Gr.M.R | 5 | 7 | 2 | **1** | 22.0 | **19.7** | 27.5 | 23.1 | 15.62 | 17.28 | 14.54 | **12.31** | 65.01 | 80.56 | 58.36 | **51.93** |
| Gr.L.R | 8 | 9 | 3 | **3** | **27.2** | 27.4 | 29.4 | 28.8 | 93.53 | 101.26 | 68.87 | **51.16** | 34.96 | 40.51 | 28.22 | **24.55** |
| M.T-I | 0 | 1 | 0 | **0** | **17.9** | 18.2 | 19.4 | 18.9 | 3.97 | 5.38 | 4.91 | **3.39** | 72.37 | 83.17 | 70.36 | **68.55** |
| M.T-II | 5 | 7 | **4** | 5 | 23.1 | **18.1** | 30.2 | 28.5 | 93.37 | 95.47 | 80.43 | **78.82** | 54.13 | 77.49 | 52.51 | **47.93** |

## 5.2 Navigating over Complex Off-road Unstructured Terrains

In the second set of experiments, we evaluate our approach when the robot navigates over complex off-road unstructured terrain. The tracks in these experiments either show transitions between different terrain types (i.e., grass to large rocks: Gr.L.R, and grass to medium rocks: Gr.M.R) or a mixture of different terrain types in real off-road environments (i.e., Mixed Terrain I: M.T-I and Mixed Terrain II: M.T-II), as shown in Fig. 4. In the experiments, no additional training is performed and the previously trained model from individual types of unstructured terrains is used directly.

Table 2 presents the quantitative results obtained by our approach and the comparison with other methods. It is observed that each of the methods have a much higher failure rate in general, especially over the M.T-II and the Gr.L.R terrains. Our approach significantly outperforms LfD and MfD in terms of failure rate. Similar to the experiments over individual types of terrain, we observe that both LfD methods have slower traversal time for successful runs, but they have a significantly higher failure rate. Moreover, LfD and MfD are outperformed with respect to inconsistency and jerkiness, with MfD performing worst among all tested methods, especially on the jerkiness metric.

Although both TRAL and our method obtain promising performance, as the main goal is to enhance consistent maneuverability, our approach obtains an average of 15.83% less on inconsistency over TRAL (with 7.66% less over traversal time, 4.50% less over failure rate, and 4.56% less on jerkiness). The p-value for inconsistency improvement is 0.004, indicating that the improvement is statistically significant. This improvement is most likely to be caused by the closed loop feedback, as it is the biggest fundamental difference between the methods.

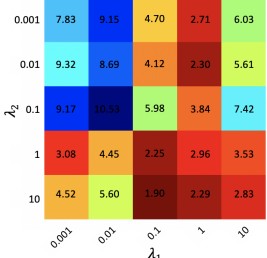

Figure 5: Hyperparameter Analysis.

## 5.3 Discussion

**Hyperparameter Analysis:** The hyperparameters $\lambda_1$ and $\lambda_2$ in Eq. (5) are implemented to balance the loss function and regularization terms. Fig. 5 depicts how the inconsistency metric changes given varying $\lambda$ values based on cross-validation during training. It is observed that $\lambda_1 \in (0.1, 10)$ and $\lambda_2 \in (1, 10)$ result in good performance in general. The best result is obtained when $\lambda_1 = 0.1$ and $\lambda_2 = 10$. These values of hyperparameters are then used for all the experiments.

**Dependence on Frame Sequences:** Our approach uses a sequence of historical frames with length $c$ to generate consistent behaviors. Fig. 6 shows the change of the inconsistency metric according to $c$. It is observed that our approach generally performs well when $c \in (15, 20)$, and we observe that the inconsistency metric is worst when either a small number ($c < 5$) or a big number ($c > 30$) is used under the sensing framerate of 30 Hz. These values can be mainly affected by the robot's speed and can differ between robotic platforms.

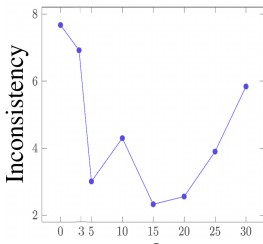

Figure 6: Seq. length.

**Convergence:** Experimental results in Fig. 7 illustrate the fast, monotonically decreasing convergence of Algorithm 1, which validates our theoretical analysis.

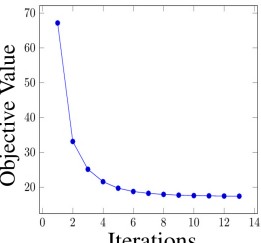

Figure 7: Convergence.

## 6 Conclusion

In this paper, we introduce a novel approach for consistent behavior generation that enables ground robots' actual behavior to more accurately match expected behaviors while adapting to a variety of unstructured off-road terrain. Our approach learns offset behaviors in a self-supervised fashion to compensate for the inconsistency between the actual and expected navigational behaviors without the need to explicitly model various setbacks, and learns the importance of the multi-modal features to improve the representation of terrain for better adaptation. Our proposed approach is extensively evaluated in real-world off-road environments. Experimental results have shown that our approach enables a robot to improve its ground maneuverability when traversing over complex unstructured off-road terrain with more behavior consistency and smoothness compared to previous methods.

## Acknowledgements

This work was partially supported by the NSF CAREER Award IIS-1942056, NSF CNS-1823245, and ARL SARA Program W911NF-20-2-0107.

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
