# OpenReview forum: "Enhancing Consistent Ground Maneuverability by Robot Adaptation to Complex Off-Road Terrains"
_robot-learning.org/CoRL/2021/Conference — CoRL2021 Oral_

### Official Review · Reviewer_7mFQ · 2021-07-19

**Originality:** Very Good
**Technical Quality:** Very Good
**Clarity Of Presentation:** Excellent
**Impact:** 2

**Recommendation:**

Weak Accept: I recommend accepting the paper, but will not argue for my recommendation if the majority of other reviewers have a different opinion.

**Summary:**

This paper investigates off-road ground robot navigation, particularly aiming to learn a control policy that provides offsets to provided vehicle commands such that the vehicle does not get stuck as often. The technical contribution is an algorithm for updating the weights of the control policy using self-supervision (i.e. learning from windows of (state, expected trajectory, actual trajectory) tuples). The authors numerically compare the proposed method with three baseline algorithms on a husky robot in various off-road terrains.

**Issues:**

Minor issues:
- The introduction repeats itself a few times
- The related work begins "Related research on robot terrain adaptation..." but this seems incomplete. The related work should better align with the paper's claimed contributions, such as a self-supervision methods, learning offset behaviors (e.g. Schoellig's work with UAVs), optimization methods with guarantees, state-of-the-art in off-road experimental validation, etc. What about the work of Kahn at Berkeley?
- " Due to momentum, we know the robot’s behaviors at the present time are dependent on previous time steps" -- ? depends on what is in the state vector
- " learning of non-linear tasks can be lifted to linear space" -- provide support for this claim?
- " for short periods of time (i.e., c = [1, 30]), the dynamics of the robot do not dramatically change and thus, the non-linearity in the robot is not severe" -- questionable...
- "behavior" should be defined more clearly as it is used throughout the paper. Is it a trajectory? or something more?
- Is this really "adaptation" as the title suggests? It's more of learning an offset policy. Adaptation suggests an online component.

**Reviewer Expertise:**

Very good: Comprehensive knowledge of the area

**Strengths And Weaknesses:**

Strengths:
- Navigating off-road is a challenging and important problem, and the authors motivate and explain the problem nicely. The paper is written well and it was enjoyable to read.
- The ability to learn in a self-supervised manner (without requiring annotation) is a great attribute of the approach
- The hardware experiments are indeed extensive: several runs of several baseline algorithms are compared on various terrains on a real hardware platform in several real off-road environments. That is a lot of work!

Weaknesses:
- It isn't obvious to me how novel the optimization algorithm is. It looks like a fairly standard regularized loss, and the authors propose to iteratively update the two weight matrices by taking the derivative and setting it to zero. If the convergence guarantee is novel then it would make sense to have it in the main text or at least a sketch of how the proof is done.
- The improvement over TRAL is rather small, and for n=10, are the differences statistically significant? "The clear performance improvement obtained by our approach is because of the closed-loop capability that considers robot states along with terrain characteristics when adapting to terrains." -- there's no ablation, so how to justify this claim?
- The quality/quantity/diversity of the training data is arguably the most important design "parameter" for the algorithm, so it would be great if the authors could provide some discussion on insights or impacts of these factors on the results.

**Summary Of Recommendation:**

The paper is solving an important and relevant problem, seems technically sound, and provides extensive hardware evaluation which can provide insights to future research in this direction. However, the technical novelty is not as clear to me, in terms of how unique the self-supervised formulation is, how unique the optimization algorithm is, and to what extent this is better than approaches such as actually modeling things like wheel slip/payload weight, off-road learning methods from Kahn et al, learning other types of offset terms (eg from Schoellig et al).

---

> ### Author Response · Authors · 2021-08-30
> **Response to the Comments by Reviewer 7mFQ**
>
> We thank the reviewer for the constructive, detailed feedback.
>
>
> Q1: Clarify the novelty of the optimization algorithm.
>
> Please see our response to Q1 of the meta-review, which clarifies the algorithm’s novelty. As a sketch, the proof is derived by alternating minimization. It alternatively updates parameter tensors U and W. In each iteration, we update W while keeping U fixed. Then we update U while keeping W fixed. The iterations continue until convergence. We have added this to Lines 203-207.
>
>
> Q2: Improve completeness of related work.
>
> We agree that a more complete review can improve the paper. We have added and compared with the references suggested (including Schoellig's and Kahn’s works) in Sec. 2.
>
>
> Q3: “Due to momentum, we know the robot’s behaviors at the present time are dependent on previous time steps" -- depends on what is the state vector?
>
> Thank you for pointing out this confusion; the term “dependent” we used in this sentence is not accurate. What we wanted to explain is: Due to momentum, a robot often has continuous motions and observations; thus, considering a history of past observations of the terrain can provide more information to generate navigational controls at the present time. This history of past observations is represented by feature instance X that encodes features from c past time steps. We have clarified this in Lines 120-122.
>
>
> Q4: Provide support for "learning of non-linear tasks can be lifted to linear space".
>
> This is supported by the Koopman operator theory, which shows that learning non-linear tasks can be lifted to a linear space by using high-dimensional data:
>
> * I. Abraham and T. D. Murphey. Active learning of dynamics for data-driven control using Koopman operators. In TRO, 35(5):1071–1083, 2019.
>
> We have added [47,48] in Line 137.
>
>
> Q5: Justify "for short periods of time (i.e., c = [1, 30]), the dynamics of the robot do not dramatically change and thus, the non-linearity in the robot is not severe"
>
> The iterative Linear Quadratic Regulator (iLQR) theory supports this claim, which solves a nonlinear problem by optimizing incremental linear problems. This theory has been used to address non-linear robotics problems, such as humanoid walking:
>
> * Y. Tassa, T. Erez, and E. Todorov. Synthesis and stabilization of complex behaviors through online trajectory optimization. In IROS, 2012.
>
> We have added [45,46] in Lines 135.
>
>
> Q6: More clearly define “behavior”.
>
> We defined behavior as control variables including linear and angular velocities that decide the robot’s motion at the present time. “Behavior” is not a trajectory in our case, as our method is a local controller that decides current velocity. We have further clarified this in Lines 118-120.
>
>
> Q7: Define "adaptation", as adaptation suggests an online component.
>
> Adaptation is defined as a robot’s ability to adjust its behaviors according to the perceived situation. Adaptation does not necessarily include an online component. This definition of adaptation is widely adopted in the areas of adaptive control and domain adaptation, without requiring an online component. [8,15,31,32,34] follow the same definition. We have further clarified this in Lines 28-31.
>
>
> Q8: Discuss impacts of training data’s quality/quantity/diversity on the results.
>
> Similar to most ML methods, more training data from more diverse scenarios (e.g., diverse terrains) often improves the results. However, training with more data increases computational cost (See response to Q2 of R#1). It is not easy to determine the impacts of data quality on the results unfortunately. First, it is difficult to estimate data quality as it is affected by many factors such as sensor type and quality (e.g., resolution and noise), feature extraction, and human/AI-agent demos. Second, the learning model is optimized based on the current quality of data. When data quality changes (e.g., by switching to better feature extractions), the model may not be applicable anymore. We have added a discussion on these in Sec. 2 of the Supp Material.
>
>
> Q9: The introduction repeats itself a few times.
>
> We have removed redundancy in the introduction.
>
>
> Q10: Further justify the improvement over TRAL.
>
> We first would like to clarify that comparisons in Tables 1 and 2 are based on the optimal models trained on the same data, where the code of TRAL is from its authors. This means our approach uses c=15 (but not 10). As the main goal is to enhance consistent maneuverability, our approach obtains an average of 15.83% less on inconsistency over TRAL (with 7.66% less on traversal time, 4.50% less on failure rate, and 4.56% less on jerkiness). The p-value for inconsistency improvement is 0.004, indicating that the improvement is statistically significant. This improvement is most likely to be caused by the closed loop feedback, as it is the biggest fundamental difference between the methods. We have clarified this in Lines 296-303.

---

> > ### Comment · Reviewer_7mFQ · 2021-09-01
> > **revisions**
> >
> > Thanks to the authors for their clear explanations which help address most of my concerns. I still recommend acceptance.

---

### Official Review · Reviewer_cgkr · 2021-07-23

**Originality:** Very Good
**Technical Quality:** Excellent
**Clarity Of Presentation:** Very Good
**Impact:** 4

**Recommendation:**

Strong Accept: I recommend accepting the paper and will argue for my recommendation even if other reviewers hold a different opinion.

**Summary:**

The core contribution of this work focuses on enabling ground robots to make consistent navigation behaviors in unstructured terrains including grass, sand, gravel, and various rocks. By taking in terrain features from the environment, their method learns to monitor the difference between the expected and actual navigation behaviors to compute an offset that reduces the difference.

**Issues:**

Incorporating a high-level diagram as mentioned in strengths/weaknesses along with text describing it might be helpful to some readers.

**Reviewer Expertise:**

Good: General knowledge of the area

**Strengths And Weaknesses:**

The paper is easy to read for a broad robotics audience. While I am not as familiar with the problem addressed in this work, the problem definition, related works, and results are discussed in a manner that brings someone from outside the community up to speed quickly and shows convincing improvements relative to prior methods. Additionally, the notation provided in the approach is helpful to interpret this work and build upon it.

It might be helpful to some readers to understand your approach at a high-level before getting into the details in section 3.2A by providing a diagram describing all the interacting tensors, inputs, outputs, critical variables and transformations. For something like object detection, that usually translates into a network architecture. However, that might be different in your case. While the paper is tight on space, the first figure is communicating a familiar to most roboticists (i.e. actual behavior deviating from expected due to unaccounted variables). Perhaps combining that figure with a high-level diagram would be beneficial.


**Summary Of Recommendation:**

The paper is well organized, mentioning relevant related work. The paper clearly defines all notation, the problem definition, and approach. The paper presents a set of unstructured terrain traversal experiments including various terrain types and complexities. Additionally, the paper shows how the proposed method outperforms prior baselines from related work, considering multiple metrics including success, consistency, and others. I think the paper would be an excellent addition to CoRL and enjoyable to readers.

---

> ### Author Response · Authors · 2021-08-30
> **Response to the Comments by Reviewer cgkr**
>
> We thank the reviewer for the appreciation of our work. Below is our response to your suggestions.
>
> Q1: Incorporate a high-level diagram along with text description.
>
> Thank you for the suggestion, and we completely agree. In the revised paper, we have added a new Fig. 2 to provide a graphic overview of our proposed approach for generating consistent navigation under the regularized optimization framework. This figure shows how terrain features, actual behaviors, estimated behaviors, parameter tensors, and other critical variables interact in our mathematical problem formulation.

---

> > ### Comment · Reviewer_cgkr · 2021-09-03
> > **reviewer response to author rebuttal**
> >
> > I would like to thank the authors for their detailed response and revisions. The authors have addressed my concerns. My score remains the same.

---

### Official Review · Reviewer_1Tow · 2021-07-23

**Originality:** Good
**Technical Quality:** Very Good
**Clarity Of Presentation:** Fair
**Impact:** 4

**Recommendation:**

Weak Accept: I recommend accepting the paper, but will not argue for my recommendation if the majority of other reviewers have a different opinion.

**Summary:**

This paper considers the problem of terrain adaptation for ground robots to traverse unstructured off-road terrains. The authors motivate the work by observing that in realistic scenarios, robots often fail to execute behaviors due to unexpected and difficult to model aspects such as wheel slip, reduction in tyre pressure etc. Hence, this paper presents a data-driven approach that learns to predict control inputs from observed features such that the robot is able to correct for unexpected observations and still consistently perform the behavior. The authors frame the problem as learning a weight tensor that associates a particular terrain feature captured at a time step with a particular behavior type. The optimization objective implicitly encodes an adaptation for correcting the behaviors online. Experiments are carried out in realistic outdoor settings on a data set captured in a certain types of unstructured terrains with features derived from multiple modalities.

**Issues:**

Issues
- I would request the authors to provide details on how the work advances the state of the art and highlight the novelty of the approach.
- The authors are also requested to explicitly state the assumptions made in the problem formulation. This would help understand the generality of the method.
- Does the approach assume that all sensor measurements are available at all times? Can the approach work when one of the sensors fails or when the sensor data arrives at different periodicities.

**Reviewer Expertise:**

Very good: Comprehensive knowledge of the area

**Strengths And Weaknesses:**

Strengths
- The authors tackle and important problem of robust execution of navigation behaviors with a data-driven approach.
- The proposed model is demonstrated on a realistic and a challenging data set.
- The authors have provided the code and a video demonstrating the results on a ground robot.

Weaknesses
- The authors rely on a certain predefined feature sets such as HOG, Locally Binary Patters etc. Is it possible to ameliorate the need for feature extraction by using a neural-style approach that can learn feature  representations.


**Summary Of Recommendation:**

Overall, the paper addresses a practical problem and provides strong results. However, the technical exposition can provide further details on how the paper advances the state of the art. Such an explanation is necessary for wider adoption of this approach. Otherwise the results are convincing.

---

> ### Author Response · Authors · 2021-08-30
> **Response to the Comments by Reviewer 1Tow**
>
> Thank you for the constructive review of our paper. We have responded to each of your comments as follows.
>
>
> Q1: Clarify how the work advances the state of the art and highlight the novelty of the approach.
>
> Please see our response to Q1 in the meta-review.
>
>
> Q2: Explicitly state the assumptions made in the problem formulation.
>
> Our approach is based on two main assumptions:
>
> The first assumption is that the LiDAR-based SLAM used in our autonomy stack (Fig. 2 in Supp Material) provides accurate estimation of the robot’s actual behavior. However, this assumption may not always be satisfied, e.g., when a robot navigates in a feature-sparse environment such as a hallway with white walls. In the case of inaccurate SLAM, the generated offset behaviors can be sub-optimal. On the other hand, our focus is the unstructured off-road environment (e.g., in a forest), which is often feature-rich. This assumption is typically satisfied. If not, we may switch to SLAM methods designed for feature-sparse scenarios or use a Visual-Inertial Navigation System (VINS) in environments with visual texture, e.g.:
>
> *  Z. Ren, L. Wang, and L. Bi. Robust GICP-based 3D LiDAR SLAM for underground mining environment. In Sensors, 19(13), p.2915, 2019.
> *  J. Zhang, M. Ren, P. Wang, J. Meng, and Y. Mu. Indoor Localization Based on VIO System and Three-Dimensional Map Matching. In Sensors, 20(10), p.2790, 2020.
>
> Because our approach works as a local controller, the second assumption is that the dynamics of the robot do not dramatically change over a shorter period of time (e.g., within one second), and the non-linearity in the robot is not severe. This assumption can be easily satisfied when we use the robot’s perception data from the past 0.5 seconds, i.e., a window size c=15 (Fig. 6).
>
> In addition, our approach also makes common assumptions such as the robot functions correctly without failures in sensing and actuation, and the expert demonstrations are reasonably good.
>
> We have discussed the assumptions in Sec. 4 of the revised Supp Material.
>
>
> Q3: Does the approach assume that all sensor measurements are available at all times? Can the approach work when one of the sensors fails or when the sensor data arrives at different periodicities
>
> Our approach discussed in this paper does not address sensor failures but assumes the robot functions correctly. Different sensor periodicities are often addressed at the robot’s hardware level, e.g., using a MasterClock in our case (which is a precise timing system to synchronize all sensor measurements).
>
> Our approach does not explicitly assume that all sensor measurements are always available, but it also does not explicitly address the problem of missing data (e.g., caused by sensor failure or occlusion). Implicitly, our approach is robust to missing data because it fuses measurements from multiple sensors to generate navigational behaviors. That is, if one sensor fails we still have information from the other sensors to generate behaviors. Each sensor contributes differently to navigational behavior generation. For example, from Fig. 3 of the Supp Material, we observe that features obtained from the color camera (HOG and LBP) have a relative importance of ~60%, and features from IMU measurements have a relative importance of ~27%. Thus, missing measurements from the color camera is more severe than missing IMU data. We have included a discussion of this point to Sec. 4 of the revised Supp Material.
>
>
> Q4: Is it possible to ameliorate the need for feature extraction by using a neural-style approach that can learn feature representations.
>
> Our approach can work with any feature extraction methods (e.g., including neural-style deep features) as long as the methods provide an informative representation of the terrain. As shown by the diagram of our autonomy stack in Fig. 1 of the Supp Material, feature extraction is a separate module in the autonomy stack. When multiple types of features are present, our approach can estimate the importance of each feature type toward generating consistent navigation behaviors (Fig. 3 of the Supp Material). We use HOG, LBP and grid-wise elevation features because they can be calculated in real time by the robot at 30 Hz without a GPU, and because they provide a good representation of the terrain. For example, HOG features encode shape, LBP features encode texture, and the elevation maps computed from LiDAR encode the 3D geometry of the terrain around the robot. Testing our approach with neural-style feature extraction methods can be an attractive future study.

---

### Official Review · Reviewer_M3C8 · 2021-07-28

**Originality:** Good
**Technical Quality:** Very Good
**Clarity Of Presentation:** Good
**Impact:** 3

**Recommendation:**

Weak Accept: I recommend accepting the paper, but will not argue for my recommendation if the majority of other reviewers have a different opinion.

**Summary:**

This paper presents a method for navigation with ground robots in unstructured outdoors environments. The method in this paper is based on learning from demonstrations, where the goal is to reproduce recorded navigation behaviors conditioned on multi-modal sensor data. To do so, the paper describes an algorithm maps 3d tensors of features extracted from sensor streams, representing a time window of past sensor readings,  to control signals through a linear operator (mode-3 tensor product). Control signals to be executed by the robot are thus computed as a linear combination of  sensor features, with weights learned from demonstrations.  The approach introduces an extra term to deal with divergence of executed robot behaviours from demonstrated behaviours, and regularization terms that ensure that the executed behaviours are temporally consistent. The paper describes an iterative optimization algorithm to solve for the weight tensors, with theoretical analysis for its convergence and an experimental comparison against other learning from demonstration methods applied to the task of navigation over various terrains with a real ground robot.

**Issues:**

The paper is not clear in a couple implementation details. For instance, is algorithm 1 executed periodically at test time to include new data points for the actual robot behaviours? or does this happen in an offline data collection phase to construct A with the same size as Y? In terms of computational cost, how does the algorithm's performance scale with varying temporal window size c, and number of demonstrated trajectories n.

Line 36 therefor
Line 162 y(t) -y(t)

**Reviewer Expertise:**

Very good: Comprehensive knowledge of the area

**Strengths And Weaknesses:**

The paper provides a  clean and simple method for learning to navigate unstructured environments from demonstrations. The paper is well written, with thorough theoretical analysis and experimental validation.

The paper could do a better job in describing the implementation details of the algorithm. While some of this information is available in the appendix, it doesn't explain very well how the training and execution phases are carried out. The video shows a failure case of the approach, but it would be better if some discussion of the limitation of the approach was added to the paper. There is no  discussion about the computational requirements and the final performance of the algorithm as a function of dataset and temporal window size, which would make this paper a stronger contribution.

**Summary Of Recommendation:**

This paper  presents an approach for solving the difficult task of unstructured navigation. The theoretical analysis and experimental validation make this a good contribution for this conference.

---

> ### Author Response · Authors · 2021-08-30
> **Response to the Comments by Reviewer M3C8**
>
> We thank the reviewer for the constructive feedback. Our response to each of your comments is provided below.
>
>
> Q1: Provide more algorithm implementation details: (1) Is Algorithm 1 executed periodically at test time to include new data points for the actual robot behaviors? or does this happen in an offline data collection phase to construct A with the same size as Y? (2) How the training and execution phases are carried out.
>
> Algorithm implementation: The optimization algorithm is executed during the offline training phase, and no online optimization is used at the test/execution phase.  For each expected behavior Y, the robot estimates its actual behavior A using a LiDAR-based SLAM method, so the dimensionality of A and Y is the same.
>
> Training/testing procedures: We train our approach on a dataset recorded while an expert (a human in our case) demonstrates robot driving over individual types of terrains, including grass, sand, gravel, medium-sized rocks, and large-sized rocks. The recorded data includes the robot’s observations from IMU, RGBD camera, and LiDAR sensors, the robot’s actual behavior, and the expected behavior demonstrated by the expert, and this dataset is used to train our algorithm and identify the optimal hyperparameter values (e.g., Figs. 5 and 6) in the training phase. The training dataset includes ~20K instances from ~5.5 hours of driving, where each instance includes the inputs from c time steps as defined in the main paper. In the testing phase, our approach uses the parameters learned during the training phase without additional online learning/optimization. Although our approach is trained using data obtained from individual types of terrains, we include unseen off-road terrains (e.g., grass-medium rocks, and mixed terrains) during testing to evaluate our approach’s ability to let ground robots generate consistent behaviors in unfamiliar terrains.
>
> In our revision, we have added more details of the algorithm implementation and the training/testing procedure in Sec. 2 of the Supp Material.
>
>
> Q2: In terms of computational cost, how does the algorithm's performance scale with varying temporal window size c, and number of demonstrated trajectories n.
>
> The computational cost increases quadratically with c in each iteration of our algorithm. The number of iterations to convergence may also increase with the increase of c (which unfortunately does not have a mathematical bound). In practice, when the algorithm is trained using the dataset on the robot’s onboard computer that has a 4.3 GHz Intel i7 processor and 16GB memory with no GPU, it takes ~3.5 hours for the algorithm to converge when c=15, and ~21 hours when c=40. We did not test the cases when c>40 because the performance has already significantly dropped when c>30 (Fig. 6). The computational cost is linearly dependent on n. We have included this discussion in Sec. 2 of the Supp Material.
>
>
> Q3: Discuss the limitation of the approach (e.g., the failure case in the video).
>
> The first limitation is that our approach is a local controller that has to work with an external local planner (shown by the autonomy stack in Fig. 1 of the Supp Material). The local planner we used does not have terrain awareness, which caused most failures in our experiments, including the failure case in the video. Another limitation is that we assume the LiDAR-based SLAM method in the robot’s autonomy can provide good estimations of the robot’s actual behavior. This assumption is not always true when the robot is in a feature-sparse environment (e.g., in a hallway with white walls). Fortunately, in practice, the SLAM method works well in off-road field environments. We have included a discussion on the limitations and assumptions of our approach in Sec. 4 of the revised Supp Material.
>
>
> Q4: Typos: Line 36 therefor, Line 162 y(t) -y(t)
>
> Thank you for pointing out these typos. We have corrected them and proofread the entire paper.

---

### Meta-Review · Area_Chair_cZZm · 2021-08-15

**Recommendation:** Accept (Oral)
**Confidence:** 4

**Metareview:**

This paper presents a method for navigation with ground robots in unstructured outdoor environments. All the reviewers agreed on the good quality of the contribution. The reviewers particularly appreciated extensive robot experiments.
- Several reviewers have pointed out the need for clearly highlighting the novel contributions of the proposed approach and how it advances the state-of-the-art.
- Providing more details about the algorithm, training procedure, and the training/test datasets can significantly strengthen the paper.
Besides addressing the major issues mentioned above, the authors should also revise the manuscript according to the other clarifications
requested and the suggestions provided.

===== Post rebuttal =====
The authors have sufficiently addressed reviewers' concerns. All reviewers agree that the paper is well written and a good contribution. I recommend an acceptance.

---

> ### Author Response · Authors · 2021-08-30
> **Response to the Comments by Area Chair cZZm**
>
> We thank the Area Chair for constructively summarizing the reviews. Below please find our responses to each comment/question (Q).
>
>
> Q1: Clearly highlight the novel contributions of the approach and how it advances the state-of-the-art.
>
> We appreciate the suggestion of highlighting the novelty and advancement of the state of the art in greater detail. In the revision, we have (1) rewritten the paragraphs in Lines 59-72 to clearly highlight the novelties and contributions, and (2) added a new Sec. 4 in the Supp Material to clearly describe the approach’s contributions and how it advances the state-of-the-art. We include the clarifications here as well:
>
> The main novelty focuses on a new tensor-based optimization formulation that formulates the real-world problem of consistent ground navigation over diverse off-road terrains into a mathematical problem. This formulation realizes our new idea of generating consistent navigation through learning offset behaviors that adapt a robot’s navigation to various unstructured terrains. Our approach continuously monitors the difference between the actual and expected navigational behaviors and computes an offset to reduce this difference. It enables consistent navigation without the requirement of explicitly modeling the setbacks that cause the behavior difference. In addition, our formulation fuses multi-modal features to enable terrain-aware ground navigation, and automatically estimates the importance of terrain features (in Supp Material Fig. 3). This ability is implemented through a structured norm over parameter tensors as a regularization term under the unified mathematical formulation.
>
> As the second novelty, our paper implements a new optimization algorithm to solve the formulated regularized optimization problem, which holds a theoretical guarantee to effectively converge to the global optimal solution. The formulated optimization problem in Eq. (5) is not easy to solve because the regularization terms are not smooth and because the objective function includes dependent variables. Since our regularization terms cannot be differentiated at non-smooth points during optimization, second-order optimization algorithms (such as Newton’s or Secant’s method) are not applicable. To address the non-smooth terms and dependent variables, we design a new alternating minimization algorithm, which can be viewed as a specialized version of gradient descent. Given a specific objective function, the solution must be mathematically derived. Although gradient descent is a fundamental mathematical tool, deriving a closed-form solution with the convergence guarantee is not always possible. Our new algorithm provides such a closed-form solution and is guaranteed to converge to the global optima. Additionally, our derived algorithm converges monotonically and fast, e.g., within tens of iterations (Fig. 7).
>
> As a practical contribution, our work provides intensive experiments and evaluations using physical robots in real-world unstructured environments. The experiments cover a set of navigation scenarios over a wide variety of individual and complex off-road terrains. We also compare with multiple previous state-of-the-art learning-based adaptation methods.
>
> This paper advances the state-of-the-art in both the problem and solution domains. In the problem domain, this paper addresses an essential but not well studied research problem, i.e., consistent ground navigation over unstructured off-road terrain. Inconsistent navigation causes slower or uncertain traversal time, which may result in mission failure when robots need to arrive at certain places on time. Inconsistent navigation may also cause errors in robot localization. This problem is essential to the success of off-road ground navigation, but has not been well addressed yet. In the solution domain, previous learning-based methods for robot navigation generally ignore behavior inconsistency caused by the setbacks. This paper advances the solution domain by introducing a novel approach with a mathematical formulation and an optimization algorithm to enable consistent ground navigation with solid mathematical support and experimental evaluation.
>
>
> Q2: Provide more details about the algorithm, training procedure, and the training/test datasets. Revise the paper according to the other clarifications and suggestions provided.
>
> To provide more details, we have (1) added a new Fig. 2 to graphically illustrate the approach and algorithm, (2) added a new Sec. 2 in the Supp Material to provide more details of algorithm implementation and training/testing procedures/datasets, and (3) added a new Sec. 4 in the Supp Material to summarize the novelty and advancements, and discuss our method’s assumptions and limitations. We included these details in our response to the corresponding questions by each reviewer. We have also revised the manuscript to address all the other comments and suggestions provided.

---

### Decision · Program_Chairs · 2021-09-13

**Decision:**

Accept (Oral)

**Comment:**

This paper presents a method for navigation with ground robots in unstructured outdoor environments. All the reviewers agreed on the good quality of the contribution. The reviewers particularly appreciated extensive robot experiments.
- Several reviewers have pointed out the need for clearly highlighting the novel contributions of the proposed approach and how it advances the state-of-the-art.
- Providing more details about the algorithm, training procedure, and the training/test datasets can significantly strengthen the paper.
Besides addressing the major issues mentioned above, the authors should also revise the manuscript according to the other clarifications
requested and the suggestions provided.

===== Post rebuttal =====
The authors have sufficiently addressed reviewers' concerns. All reviewers agree that the paper is well written and a good contribution. I recommend an acceptance.